# A Quadruplex RT-qPCR for the Detection of African Swine Fever Virus, Classical Swine Fever Virus, Porcine Reproductive and Respiratory Syndrome Virus, and Porcine Pseudorabies Virus

**DOI:** 10.3390/ani14233551

**Published:** 2024-12-09

**Authors:** Zhuo Feng, Kaichuang Shi, Yanwen Yin, Yuwen Shi, Shuping Feng, Feng Long, Zuzhang Wei, Hongbin Si

**Affiliations:** 1College of Animal Science and Technology, Guangxi University, Nanning 530005, China; fz2187941887@163.com (Z.F.); shiyuwen2@126.com (Y.S.); shb2009@gxu.edu.cn (H.S.); 2Guangxi Center for Animal Disease Control and Prevention, Nanning 530001, China; yanwen0349@126.com (Y.Y.); fsp166@163.com (S.F.); longfeng1136@163.com (F.L.)

**Keywords:** African swine fever virus (ASFV), porcine reproductive and respiratory syndrome virus (PRRSV), classical swine fever virus (CSFV), porcine pseudorabies virus (PRV), multiplex RT-qPCR

## Abstract

African swine fever virus (ASFV), porcine reproductive and respiratory syndrome virus (PRRSV), classical swine fever virus (CSFV), and porcine pseudorabies virus (PRV) are important pathogens circulating in pig herds in many countries. Pigs infected with these viruses show similar signs, such as increased body temperature and loss of appetite, and are prone to respiratory symptoms, digestive symptoms, neurological symptoms, and/or abortion in pregnant sows. To accurately detect these viruses and differentially diagnose these diseases, four pairs of specific primers and TaqMan probes were designed aiming at the *B646L* (p72) gene of ASFV, the 5′ untranslated region (*5′UTR*) of CSFV, the *ORF6* gene of PRRSV, and the *gB* gene of PRV. After optimizing the reaction conditions, a multiplex real-time quantitative RT-PCR (RT-qPCR) assay was developed for the differential detection of ASFV, CSFV, PRRSV, and PRV. The assay was further validated to test 3116 clinical samples, and the positivity rates of ASFV, CSFV, PRRSV, and PRV were 10.84% (338/3116), 0.80% (25/3116), 14.92% (465/3116), and 1.38% (43/3116), respectively. The assay provides a highly specific, sensitive, and reproducible method for the detection of ASFV, CSFV, PRRSV, and PRV.

## 1. Introduction

African swine fever virus (ASFV), porcine reproductive and respiratory syndrome virus (PRRSV), classical swine fever virus (CSFV), and porcine pseudorabies virus (PRV) are important swine pathogens that cause serious harm and great economic losses to the pig industry around the world. All four viruses cause a series of respiratory, digestive, neurological, and/or reproductive symptoms, which are similar and not easy to distinguish.

African swine fever (ASF) is a highly transmissible animal disease with high mortality and morbidity. The causative agent, ASFV, was first described in Kenya in 1921 and subsequently spread to other countries in Africa, Europe, Asia, and America [1]. ASFV was first reported in China on 3 August 2018 and then spread to most provinces around the country, endangering China’s pig industry and causing substantial economic losses [2,3]. ASFV is classified into the *Asfarviridae* family and is a double-stranded DNA virus with a genome size of 170–190 kb [1]. Pigs infected with different ASFV strains show different clinical signs and pathological lesions. Highly pathogenic strains cause high fever, extensive hemorrhage accompanied by a series of respiratory and gastrointestinal symptoms, and congestive enlargement of the spleen [4,5]. To date, all strains reported outside Africa are genotypes I and II [6]. Both genotype I and II strains are circulating in pig herds in the Guangxi province of China [7,8].

Classical swine fever (CSF) was initially identified in 1810 in the United States. Since the 1960s, CSF has progressively disseminated in many countries [9]. CSFV, the etiological agent of CSF, is categorized within the genus *Pestivirus* of the *Flaviviridae* family. It is a single-stranded, positive-sense RNA virus with a genome size of 12.3 kb, and the genome encodes four structural proteins and eight non-structural proteins [10]. CSF was first identified in China in 1920 [11]. The main symptoms of CSF are high fever, skin, and mucous hemorrhage accompanied by gastrointestinal symptoms, pathological features such as bleeding from the lymph nodes, spleen, kidneys, and gastrointestinal tract, and, in severe cases, neurological symptoms, with a high lethality rate [12]. CSF has been successfully controlled with mandatory immunization with the live attenuated CSF vaccine in many countries. However, CSF has not been completely eradicated in China until now [13,14,15].

Porcine reproductive and respiratory syndrome (PRRS) emerged in the 1980s and has caused huge economic damage to the global pig industry [16,17]. In addition to atypical symptoms such as depression, fever, and skin and mucous hemorrhage, it also causes miscarriage in pregnant sows and respiratory diseases in pigs of all ages, demonstrating a high mortality rate [18]. PRRSV is categorized within the genus *Betaarterivirus* of the family *Arteriviridae*, with positive-sense RNA of 12.7–15.7 kb genome size. PRRSV is divided into two species: *Betaarterivirus suid* 1 (formerly designated as PRRSV-1 or European PRRSV) and *Betaarterivirus suid* 2 (formerly designated as PRRSV-2 or North American PRRSV) [19,20]. PRRSV-2 was first identified in China in 1996, and at present, both PRRSV-1 and PRRSV-2 are circulating in China, with PRRSV-2 predominant and PRRSV-1 sporadic [17,21]. PRRSV disrupts the host’s immune responses, eludes viral clearance by the host, and exhibits a rapid mutation rate, complicating prevention and control efforts [22].

PRV, the causative agent of pseudorabies (PR), is an enveloped double-stranded DNA virus classified in the subfamily *Alphaherpesvirinae* of the *Herpesviridae* family, with a genome size of 143 kb [23]. PRV was first identified in Hungary in 1902 and has continued to circulate in pig herds in many countries until now, bringing great harm to the global livestock industry [24,25]. PRV can infect pigs of various ages and exhibit diverse symptoms. Neonatal piglets primarily exhibit neurological signs and experience a high mortality rate, whilst infected adult sows are prone to reproductive and respiratory diseases [26]. PRV not only infects pigs; it can also spread across species and infect other species including humans, mammals, carnivores, and rodents, and there have been cases of acute encephalitis in humans [27,28,29]. Due to the mutation of PRV, the vaccine cannot provide complete protection, and this disease is still prevalent in China and other countries at present [15,24,30]. Therefore, PRV not only jeopardizes agricultural production but also poses a threat to human health.

ASFV, CSFV, PRRSV, and PRV are still circulating in pig herds in China and other countries, and they cause similar clinical manifestations and pathological damages, which challenge the accuracy of their differential diagnosis depending only on clinical signs and gross pathological changes. The specific, sensitive, and accurate detection of the causative agents in the laboratory is necessary. Real-time quantitative PCR (qPCR) facilitates the simultaneous detection of numerous pathogens with great sensitivity, exceptional specificity, and operational simplicity, rendering it ideal for the detection and identification of multiple pathogens [31,32]. Singleplex qPCR/RT-qPCR assays for the detection of ASFV, PRRSV, CSFV, or PRV have been reported [33,34,35,36]. These assays use one pair of specific primers and one probe to detect one virus of ASFV, PRRSV, CSFV, or PRV. Multiplex qPCR/RT-qPCR assays have also been reported to concurrently test two or three of the aforementioned four viruses [15,37,38,39]. These assays use two or three pairs of specific primers and corresponding probes to detect two or three viruses of ASFV, PRRSV, CSFV, and PRV, i.e., CSFV/PRV [15], ASFV/CSFV [38], PRRSV/PRV [39], and ASFV/CSFV/PRRSV [37]. However, no multiplex RT-qPCR assay capable of the simultaneous detection of ASFV, CSFV, PRRSV, and PRV has been reported until now. This study developed a one-step quadruplex TaqMan RT-qPCR assay for the rapid and specifically differential detection of ASFV, PRRSV, CSFV, and PRV. The assay offers a sensitive and accurate method for simultaneously detecting these pathogens in one reaction at the same time and performing an epidemiological investigation of these diseases in a very short time. In addition, the developed quadruplex RT-qPCR included two RNA viruses and two DNA viruses, and one-step RT-qPCR was used to avoid the separation of reverse-transcription and amplification steps and improve the assay’s amplification efficiency.

## 2. Materials and Methods

### 2.1. Reference Strains

The vaccine strains of CSFV (C strain), PRRSV (Ch-1R strain), and PRV (Bartha-K61 strain) were purchased from Huapai Biological Co., Ltd. (Chengdu, China). The positive clinical samples of genotype I ASFV and genotype II ASFV were provided by our laboratory. These vaccine strains and positive samples were used to construct the recombinant standard plasmid constructs and used as the positive controls in the developed assay’s specificity analysis.

The vaccine strains were obtained as follows: Japanese encephalitis virus (JEV, SA14-14-2 strain), H1N1 subtype swine influenza virus (SIV, TJ strain), transmissible gastroenteritis virus (TGEV, H strain), porcine epidemic diarrhea virus (PEDV, CV777 strain), porcine rotavirus (PoRV, NX strain), and porcine circovirus type 2 (PCV2, WH strain) from Huapai Biological Co., Ltd. (Chengdu, China); PRRSV (HuN4-F112 strains) from Harvac Biotechnology Co., Ltd. (Harbin, China); PRV (HN1201 strain) and CSFV (WH-09 and CVCC AV1412 strains) from Keqian Biology Co., Ltd. (Wuhan, China); PRV (HB-98 and HB2000 strains), and PRRSV (R98 strain) from China Animal Husbandry Industry Co., Ltd. (Chengdu, China); and foot-and-mouth disease virus (FMDV, Re-O/MYA98/JSCZ/2013, and Re-A/WH/09 strains) from Shenlian Biological Co., Ltd. (Shanghai, China). The positive clinical samples of PRRSV-1, PRRSV-2, porcine respiratory coronavirus (PRCoV), porcine hemagglutinating encephalomyelitis virus (PHEV), porcine deltacoronavirus (PDCoV), senecavirus A (SVA), atypical porcine pestivirus (APPV), and PCV3 were provided by our laboratory. These vaccine strains and positive samples were used as controls in the developed assay’s specificity analysis. The GenBank accession number of the abovementioned vaccine strains is provided in Appendix A.

### 2.2. Clinical Samples

A total of 3116 clinical samples, including lymph node, lung, tonsil, kidney, and spleen from each pig (all tissues from each pig were homogenized, and each pig’s homogenized tissue was considered as one sample for detection of viruses), were collected from pig farms, slaughterhouses, and harmless treatment plants in Guangxi Province, China, from April 2023 to September 2024. Of the 3116 samples, 369 samples were collected from pig farms, 1784 samples from slaughterhouses, and 963 samples from harmless treatment plants. All specimens were kept at −80 °C until use.

### 2.3. Primers and Probes

Based on the sequences of 51 ASFV strains, 33 CSFV strains, 31 PRRSV strains, and 40 PRV strains available in the NCBI GenBank (https://www.ncbi.nlm.nih.gov/nucleotide/ (accessed on 15 October 2022)) (Appendix A), multiple sequence alignments were performed using Clustal W of DNAstar Ver 6.0 (https://www.dnastar.com/ (accessed on 15 October 2022)), and the conserved regions were used to design primers and probes using Oligo software (Version 7.60) (https://www.oligo.net/ (accessed on 15 October 2022)) (Appendix A). The primers and TaqMan probes targeted the CSFV *5′UTR* gene and the PRRSV *ORF6* gene (Table 1). The primers and probes for ASFV and PRV targeted the *B646L* gene and the PRV *gB* gene, which were used and validated in previous reports [8,39]. The primers and probe for ASFV can detect 24 genotypes of ASFV [8]. The primers and probes were synthesized by TaKaRa Biotechnology Co., Ltd. (Dalian, China).

### 2.4. Total Nucleic Acid

The tissue specimens of lymph nodes, lungs, tonsils, kidneys, and spleens (about 0.1 g each tissue) were put into a 2.0 mL EP tube, followed by the addition of phosphate buffer solution (PBS, pH7.2, *w*/*v* = 1:3). The tissues were frozen and thawed three times; then, they were homogenized using a Retsch tissue homogenizer (Haan, Germany) with 30 Hz and centrifuged at 12,000× *g* for 5 min at 4 °C. The 200 μL tissue supernatant, and the vaccine solution were used to extract nucleic acid using the Tianlong Viral RNA/DNA Extraction Kit Ver.4.0 (Xi’an, China) and then promptly used to detect ASFV, CSFV, PRRSV, and PRV or stored at −80 °C until use.

### 2.5. Standard Plasmid Constructs

The viral RNAs/DNAs of ASFV, CSFV, PRRSV, and PRV were extracted from positive clinical samples or vaccine solutions. RNAs were reverse transcribed into cDNA. The DNA/cDNAs were used to amplify the target gene fragments of ASFV, CSFV, PRRSV, and PRV using PCR with the specific primers in Table 1. The PCR products were purified and used to construct the standard plasmids according to the previous report by Chen et al. [37]. The recombinant plasmids, designated p-ASFV, p-CSFV, p-PRRSV, and p-PRV, were used as standard plasmid constructs for the development of multiplex RT-qPCR. The OD_260_/OD_280_ nm values of the four standard plasmid constructs were obtained spectrophotometrically, and their concentrations were determined per the following formula:$$\mathrm{plasmid}\;(\mathrm{copies}/µL)=\frac{\left(6.02\times 10^{23}\right)\times (X\mathrm{ng}/µL\times 10^{-9})}{\mathrm{plasmid}\;\mathrm{length}\;(\mathrm{bp})\times 660}.$$

### 2.6. Optimal Reaction Parameters

The standard plasmid constructs p-ASFV, p-CSFV, p-PRRSV, and p-PRV were used to develop the quadruplex RT-qPCR in this study. The reaction volume of the multiplex RT-qPCR was 20 μL. The reagents used were purchased from TaKaRa Biotechnology Co., Ltd. (Dalian, China): 10 μL of 2× One-Step RT-qPCR Buffer III, 0.4 μL of Ex Taq HS (5 U/μL), 0.4 μL of Prime-Script RT Enzyme Mix II, 0.2–0.8 μL of primers and probes (20 pmol/μL), 2.0 μL of the mixed standard plasmid constructs, and 3.5 μL of distilled water. Different annealing temperatures (54, 55, 56, 57, 58, 59, 60, 61, and 62 °C), and primer and probe concentrations (0.2, 0.3, 0.4, 0.5, 0.6, 0.7, and 0.8 pmol/μL) were used for amplification with different combinations to obtain the optimal reaction conditions. The reaction conditions were optimized using an ABI QuantStudio™ 6 Real-Time System (Carlsbad, CA, USA) using the following reaction steps: 5 min at 42 °C, 10 s at 95 °C, 40 cycles of 5 s at 95 °C, and 54 °C to 62 °C for 30 s. The fluorescence signals were recorded at the end of each cycle to derive the maximum ΔRn and minimum cycling threshold (Ct) values post-amplification. The optimal annealing temperature and the optimal primer and probe concentrations were determined by the checkerboard method.

### 2.7. Standard Curves

The p-ASFV, p-CSFV, p-PRRSV, and p-PRV standard plasmid constructs mixed at an equal volume were sequentially ten-fold diluted to the final concentration from 10^8^ to 10^2^ copies/μL and used to perform qPCR for plotting the developed assay’s standard curves.

### 2.8. Analytical Specificity

The viral DNAs/RNAs of ASFV, CSFV, PRRSV, PRV, PoRV, PDCoV, TGEV, PHEV, JEV, PEDV, FMDV, SVA, SIV, PCV2, PCV3, PRCoV, and APPV were used as templates to evaluate the developed assay’s specificity. The different strains of the same target viruses (ASFV, CSFV, PRRSV, PRV) were used for inclusivity analysis, and the other swine viruses (PoRV, PDCoV, TGEV, PHEV, JEV, PEDV, FMDV, SVA, SIV, PCV2, PCV3, PRCoV, and APPV) were used for strict specificity analysis. The positive clinical samples were used as positive controls, and the negative clinical samples and nuclease-free distilled water were used as negative controls.

### 2.9. Analytical Sensitivity

The p-ASFV, p-CSFV, p-PRRSV, and p-PRV standard plasmid constructs mixed at an equal volume were sequentially ten-fold diluted to a final concentration from 10^8^ to 10^−1^ copies/μL to evaluate the developed assay’s sensitivity.

In addition, a Probit regression analysis was also used to assess the sensitivity of the assay. The mixture of the four plasmid constructs p-ASFV, p-CSFV, p-PRRSV, and p-PRV were 2-fold sequentially diluted from 500, 250, 125, to 62.5 copies/reaction and used as templates to evaluate the developed assay’s sensitivity. Based on the experimental design and statistical requirements, a sample size of 30 was selected to reach the requirement of experimental feasibility.

### 2.10. Reproducibility Analysis

The standard plasmid constructs mixed in equal proportions were ten-fold diluted to a final concentration of 10^7^, 10^5^, and 10^3^ copies/μL. To evaluate the developed assay’s reproducibility, the experiments were repeated on three different dates, and three sets of replicates were performed for each operation. The inter-assay and intra-assay variabilities were assessed by coefficients of variation (CVs).

### 2.11. Test of the Clinical Specimens

Total viral DNAs/RNAs were extracted from the 3116 clinical samples from Guangxi province during 2023–2024 using the Tianlong Viral RNA/DNA Extraction Kit (Xi’an, China) and used to detect ASFV, CSFV, PRRSV, and PRV by the developed assay in order to evaluate its clinical applicability.

In addition, the 3116 clinical samples were tested via the qPCR/RT-qPCR outlined in the World Organisation for Animal Health (WOAH)’s Terrestrial Manual 2024 for ASFV (https://www.woah.org/fileadmin/Home/eng/Health_standards/tahm/3.09.01_ASF.pdf (accessed on 23 April 2024)), Terrestrial Manual 2022 for CSFV (https://www.woah.org/fileadmin/Home/eng/Health_standards/tahm/3.09.02_CSF.pdf (accessed on 23 April 2024)), Terrestrial Manual 2018 for PRV (https://www.woah.org/fileadmin/Home/eng/Health_standards/tahm/3.01.02_AUJESZKYS.pdf (accessed on 23 April 2024)), and the RT-qPCR outlined in the Chinese entry–exit inspection and quarantine industry standard for PRRSV (https://hbba.sacinfo.org.cn/attachment/onlineRead/38852e81da9ce4d6f37bdaab97497d450348b72e1303eddbbf9be8df638c37e0 (accessed on 23 April 2024)).

In addition, the tested results of the 3116 clinical samples using the developed quadruplex RT-qPCR assay were compared with those of the reference assays. The diagnostic sensitivity and specificity were evaluated, and the coincidence rates were determined.

## 3. Results

### 3.1. Construction of the Standard Plasmids

The PCR products of the ASFV *B646L* (p72) gene, CSFV *5′UTR* gene, PRRSV *ORF6* gene, and PRV *gB* gene were used to construct the standard plasmid constructs, which were named p-ASFV, p-CSFV, p-PRRSV, and p-PRV. The initial concentrations of these four plasmids were 7.32 × 10^10^, 5.81 × 10^10^, 5.92 × 10^10^, and 6.36 × 10^10^ copies/µL, respectively, which were then diluted to 10^10^ copies/μL and kept at −80 °C until use.

### 3.2. Determination of the Optimal Parameters

The reaction system is composed of a primer, a probe, an enzyme, and nuclease-free distilled water. To optimize the reaction conditions, different concentrations of primers and probes, different annealing temperatures, and different reaction cycles were combined to perform the tests. Finally, the optimal concentrations, temperatures, and cycles were determined. After optimization of the reaction conditions, the ideal quadruplex RT-qPCR parameters were obtained. The 20 µL system and the validated multiplex RT-qPCR parameters are shown in Table 2. The optimal annealing temperature was 55 °C. The reaction procedure was as follows: 5 min at 42 °C, 10 s at 95 °C, 40 cycles of 5 s at 95 °C, and 55 °C for 30 s. The fluorescence signals were automatically obtained at the end of each cycle. The samples with Ct values of ≤36 were considered as positive samples.

### 3.3. Generation of the Standard Curves

The final concentrations of the four standard plasmid constructs ranged from 10^8^ to 10^2^ copies/μL and were used to generate the standard curves. The results indicated that the slope, R^2^, and Eff% were −3.135, 1.000, and 108.556% for ASFV; −3.142, 1.000, and 108.268% for CSFV; −3.043, 0.999, and 111.520% for PRRSV; and −3.208, 0.999, and 106.945% for PRV, respectively (Figure 1), demonstrating a strong linear correlation between the initial templates and Ct values.

### 3.4. Specificity Analysis

The RNAs/DNAs of ASFV, CSFV, PRRSV, PRV, PoRV, PDCoV, TGEV, PHEV, JEV, PEDV, FMDV, SVA, SIV, PCV2, PCV3, PRCoV, and APPV were used as templates to validate the assay’s specificity. The results indicated that the assay produced distinct amplification curves only for ASFV, CSFV, PRRSV, and PRV, while the other 13 porcine viruses did not generate fluorescence signals, demonstrating the assay’s excellent specificity (Figure 2).

### 3.5. Sensitivity Analysis

The final reaction concentrations of the four plasmid constructs in their mixture ranged from 10^8^ to 10^−1^ copies/μL and were used to assess the assay’s sensitivity. The results showed that the LODs of all four plasmid constructs were 10^1^ copies/μL, indicating great sensitivity of the assay (Figure 3).

In addition, the plasmid constructs of p-ASFV, p-CSFV, p-PRRSV, and p-PRV were two-fold sequentially diluted from 500 to 62.5 copies/reaction to assess the assay’s sensitivity using a Probit regression analysis. The Ct values and hit rates are shown in Table 3. The results indicated that the LODs for p-ASFV, p-CSFV, p-PRRSV, and p-PRV were determined to be 134.585 (95% confidence interval (CI) of 123.221–154.308), 139.831 (95% CI of 128.263–162.332), 147.076 (95% CI of 134.301–178.186), and 142.331 (95% CI of 130.476–167.081) copies/reaction, respectively (Figure 4).

### 3.6. Reproducibility

Reproducibility was evaluated using a combination of four standard plasmid constructs at final concentrations of 10^7^, 10^5^, and 10^3^ copies/μL. The results demonstrated intra-assay CVs of 0.06–0.46% and inter-assay CVs of 0.42–1.12% (Table 4), indicating the assay’s excellent reproducibility.

### 3.7. Assessment of Clinical Specimens

The 3116 clinical samples from 14 cities in Guangxi province were analyzed using the developed assay (Table 5). The positivity rates for ASFV, CSFV, PRRSV, and PRV were 10.85% (338/3116), 0.80% (25/3116), 14.92% (465/3116), and 1.38% (43/3116), respectively. The co-infection positivity rates for ASFV + CSFV + PRRSV, ASFV + PRRSV + PRV, ASFV + CSFV, ASFV + PRRSV, ASFV + PRV, PRRSV + PRV, and CSFV + PRRSV were 0.03% (1/3116), 0.06% (2/3116), 0.06% (2/3116), 1.12% (35/3116), 0.13% (4/3116), 0.26% (8/3116), and 0.19% (6/3116), respectively.

Among the 14 cities, Liuzhou and Yulin had higher positivity rates for ASFV, CSFV, PRRSV, and PRV, and Yulin had the highest total number of co-infected samples (Table 5). The samples from slaughterhouses, pig farms, and harmless treatment plants were positive for ASFV (0.54%, 4.37%, and 26.79%), CSFV (1.63%, 0.06%, and 1.87%), PRRSV (13.28%, 12.84%, and 19.42%), and PRV (2.98%, 0.22%, and 2.91%), respectively. Of the three different originations, the samples from harmless treatment plants had the highest positivity rates of ASFV, CSFV, PRRSV, and PRV. No co-infection was detected in the 1784 clinical samples from slaughterhouses (Table 6). Moreover, some positive samples were selected to amplify and sequence to verify their confirmation as ASFV, CSFV, PRRSV, and PRV.

In addition, the 3116 clinical samples were analyzed using the reference qPCR. The results indicated that the positivity rates for ASFV, CSFV, PRRSV, and PRV were 10.69% (333/3116), 0.77% (24/3116), 14.63% (456/3116), and 1.28% (40/3116), respectively. In comparison to the reference qPCR assays, the diagnostic sensitivity and specificity of the developed assay were 99.10% and 99.71% for ASFV, 95.83% and 99.94% for CSFV, 99.12% and 99.51% for PRRSV, and 97.50% and 99.87% for PRV, respectively (Table 7). The coincidence rates between the developed and reference methods surpassed 99.45% (Table 8).

## 4. Discussion

ASFV, CSFV, PRRSV, and PRV have long been causing huge threats to China’s pig industry. ASFV originated in Africa and then spread to Asia, Europe, Oceania, and America, and ASF has caused serious economic losses to the global pig industry. ASFV has a wide range of transmission routes and easily survives, and there is no commercial vaccine available [40]. In recent years, mutations and recombination of ASFV have occurred, making it more difficult to prevent and control [41,42]. From the first identification of CSF in 1810 to the subsequent outbreaks in many countries, CSF has been controlled by the application of effective vaccines [9,13,14]. Despite the eradication of CSF in many countries, it remains endemic in certain countries in Asia and the Americas. CSF is now effectively controlled in China but still not completely eradicated [11,13,14,15]. The epidemiology of PRRSV in China is complex. PRRSV-2 first appeared in 1996, PRRSV-1 was first identified in 2011 [20,43], and PRRSV-1 and PRRSV-2 co-exist in China at present [17,20]. Currently, PRRSV-2 is the predominant strain prevalent in China, mainly including HP-PRRSV, NADC34-like, and NADC30-like strains [44,45,46,47]. PRRSV vaccines have been used widely, but the complications of circulating viruses and the viral mutation and recombination have hindered the efficacy of the vaccines [48,49]. PRV can infect pigs of various ages and cause diverse clinical signs. Furthermore, PRV can be spread between species, with documented instances of pig-to-human transmission resulting in acute encephalitis [27,28,29]. In recent years, the mutations and recombination also prevent PRV vaccines from offering complete protection, resulting in the continued prevalence of PRV in China [23,30,50,51].

Infections with these viruses cause similar clinical symptoms, such as elevated body temperature, respiratory distress, vomiting, diarrhea, abortions in pregnant sows, and so on [5,12,18,26]. In recent years, due to mutation and recombination, the emergence of some low-virulence strains has caused infected pigs to show non-specific clinical signs, making them more difficult to differentiate and diagnose [41,42,49]. An accurate and reliable diagnosis depends on laboratory detection. It is necessary to establish a rapid and precise approach capable of simultaneous detection and distinguishing these four viruses. qPCR has been extensively utilized in laboratories owing to its exceptional specificity, sensitivity, and user-friendliness [31,32]. To date, qPCR/RT-qPCR assays that could detect one to three ASFV, CSFV, PRRSV, and/or PRV have been reported [33,34,35,36,37,38,39]. In this study, four pairs of specific primers and probes were designed targeting the ASFV *B646L* gene, CSFV *5′UTR* gene, PRRSV *ORF6* gene, and PRV *gB* gene (the primers and probes of ASFV and CSFV have been reported in previous papers [8,39]). After the optimization of the reaction conditions, the developed quadruplex RT-qPCR showed that the LODs of the four viruses were 10^1^ copies/μL, the intra- and inter-assay CVs were ≤1.12%, and only ASFV, CSFV, PRRSV, and PRV yielded positive fluorescent signals, demonstrating the assay’s high sensitivity, specificity, and reproducibility. Compared to the reference singleplex RT-qPCR assays, which had LODs of 10^1^ to 10^2^ copies/μL, the developed quadruplex RT-qPCR in this study had lower or similar LODs for sensitivity. In addition, the developed assay and the reference assays showed excellent specificity. The approach accomplishes the simultaneous detection of four viruses in a single tube using one reaction system and had coincidence rates higher than 99.45% with the reference assays. The assay was further used to test 3116 clinical samples to validate its applicability. All these results illustrated that the developed assay is appropriate for various laboratories to detect ASFV, CSFV, PRRSV, and PRV in clinical specimens. It is noteworthy that some positive samples tested by the developed quadruplex RT-qPCR assay were determined as negative samples when tested by the reference assays, which indicated that the developed assay was more sensitive and accurate than the reference assays for the detection of the clinical samples.

In this study, the 3116 clinical samples were tested using the developed assay, and the positivity rates for ASFV of 10.84%, PRRSV of 14.92%, CSFV of 0.80%, and PRV of 1.38% were found in Guangxi province, indicating that these viruses are still circulating in some pig herds. Particularly, PRRSV and ASFV had positivity rates higher than 10%, and this should be paid more attention due to their harm to the immune system, high morbidity, and mortality rate.

ASFV, CSFV, PRRSV, and PRV are still circulating in some pig herds. Since the first report of ASF in 2018, ASFV variant strains have been found in China. The recombinant strains of type I and type II ASFV showed high lethality and transmissibility [42]. On the other hand, the genotype II ASFV variants with decreased pathogenicity were identified in China in 2020, and genotype I ASFV with low virulence appeared in pigs in China in 2021 [52,53]. To date, the data on the prevalence of ASFV in China are very limited. Our previous investigation indicated that the ASFV positivity rate in Guangxi province during 2019–2023 was 8.16% (287/3519) [8]. In this study, the positivity rate of ASFV was 10.84%. Although the CSF vaccine (C strain) has been used widely, the vaccine does not achieve complete protection against CSFV. In 2011, China experienced 285 outbreaks of CSF in 12 provinces [11]. The CSFV positivity rate in Guangxi province from 2018 to 2021 was 9.36% (107/1143), and the positivity rate in aborted piglets in Hunan province during 2019–2021 was 12.29% (50/407) [38,54]. It is noteworthy that in 2018, CSF re-emerged in Japan after a 26-year absence, proving that countries that have eradicated CSF are still at risk of a re-endemic [55,56]. The positivity rate of PRRSV in 21 provinces in China from 2021 to 2022 was 32.1% (2416/7518), and the infections were of diverse strains, of which the dominant strain is lineage 1 [45]. The PRRSV positivity rate was 18.82% (1279/6795) in South China from 2017 to 2021 [57] and 10.17% (499/4909) in Guangxi province from 2022 to 2023 [39]. In this study, PRRSV had a positivity rate as high as 14.92%. Due to the failure of protection from the gE gene-deleted attenuated vaccine, PR has been outbroken in some pig herds in China since 2011 [23]. Between December 2017 and May 2021, the average PRV positivity rate was 25.04% in 14 provinces across the country [30]. Our previous investigation indicated that the PRV positivity rate in Guangxi province during 2022–2023 was 0.84% (41/4909) [38]. In this study, PRV had a positivity rate of 1.38%.

These above results indicated that ASFV, CSFV, PRRSV, and PRV, the four important pathogens responsible for serious animal diseases, have caused substantial economic losses in China’s pig industry. Since no commercial vaccine is available for ASFV, and the commercial vaccines used for CSFV, PRRSV, and PRV cannot provide complete protection, accurate and reliable detection methods, continuous surveillance, and thorough epidemiological studies are vital and necessary for the prevention and control of these diseases. The developed quadruplex RT-qPCR in this study meets the need for the sensitive, rapid, accurate, and reliable detection of ASFV, CSFV, PRRSV, and PRV.

## 5. Conclusions

ASF, CSF, PRRS, and PR are important swine illnesses that result in substantial economic losses to the global pig industry, especially in China. This study developed a sensitive, specific, and accurate quadruplex RT-qPCR assay for the simultaneous detection of ASFV, CSFV, PRRSV, and PRV, enabling the concurrent detection and differentiation of these viruses in a single reaction. Furthermore, ASFV, CSFV, PRRSV, and PRV remain prevalent in certain pig herds in Guangxi province, southern China.

## Figures and Tables

**Figure 1 animals-14-03551-f001:**
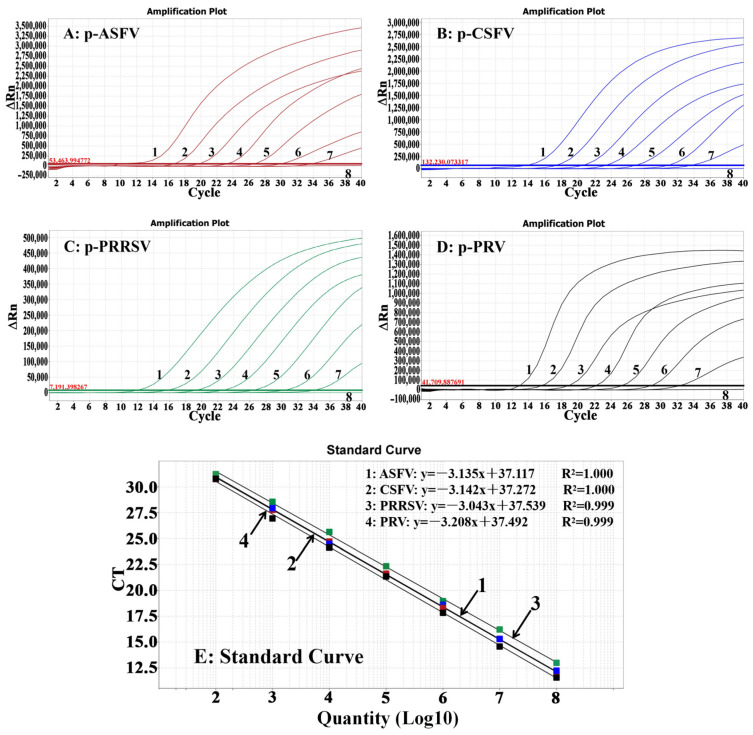
The amplification curves of p-ASFV (**A**), p-CSFV (**B**), p-PRRSV (**C**), and p-PRV (**D**) and the standard curves (**E**). In (**A**–**E**), 1–7: the concentrations of the plasmid constructs ranged from 10^8^ to 10^2^ copies/μL; 8: negative control.

**Figure 2 animals-14-03551-f002:**
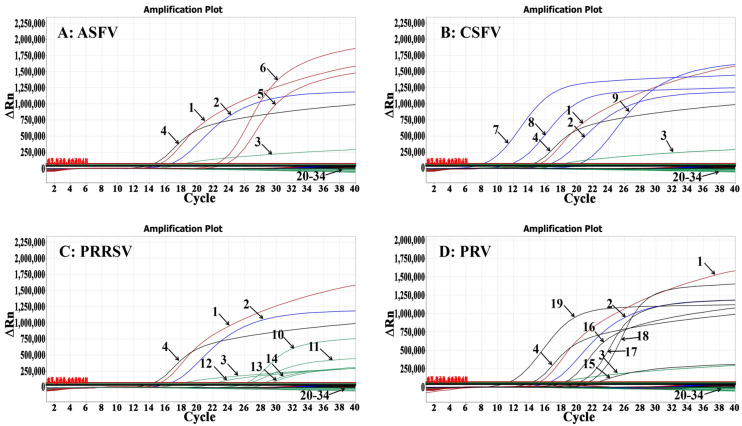
Specificity assessment of the quadruplex RT-qPCR for ASFV (**A**), CSFV (**B**), PRRSV (**C**), and PRV (**D**). In (**A**–**D**), 1: p-ASFV; 2: p-CSFV; 3: p-PRRSV; 4: p-PRV; 5–6: the positive clinical samples of genotype II ASFV and genotype I ASFV; 7: the positive clinical sample of CSFV; 8–9: CSFV CVCC AV1412, and WH-09 vaccine strains; 10–11: the positive clinical samples of PRRSV-2 and PRRSV-1; 12–14: PRRSV-2 CH-1R, HuN4-F112, and R98 vaccine strains; 15: the positive clinical sample of PRV; 16–19: PRV Bartha-k61, HB-98, HB2000, and HN1201 vaccine strains; 20: JEV SA14-14-2 vaccine strain; 21: PEDV CV777 vaccine strain; 22: TGEV H vaccine strain; 23: PoRV NX vaccine strain; 24: PCV2 WH vaccine strain; 25: SIV TJ vaccine strain; 26: FMDV Re-O/MYA98/JSCZ/2013 and Re-A/WH/09 vaccine strains; 27–32: the positive clinical samples of PRCoV, PHEV, PDCoV, PCV3, SVA, and APPV; 33: negative clinical sample; and 34: distilled water.

**Figure 3 animals-14-03551-f003:**
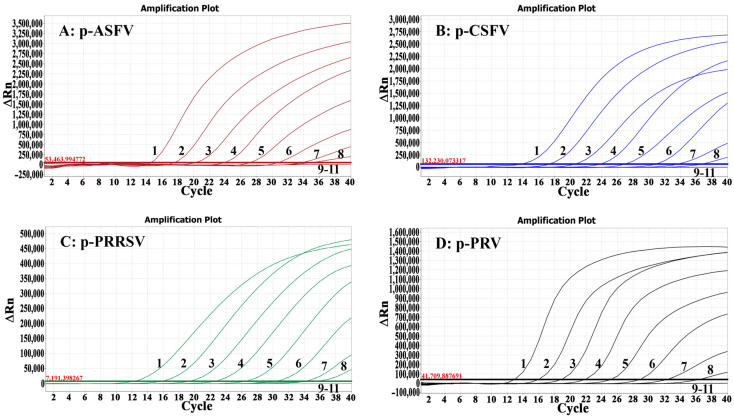
Sensitivity assessment of the quadruplex RT-qPCR. The amplification curves of p-ASFV (**A**), p-CSFV (**B**), p-PRRSV (**C**), and p-PRV (**D**) are shown. In (**A**–**D**), 1–10: the final reaction concentrations ranged from 10^8^ to 10^−1^ copies/μL; 11: distilled water.

**Figure 4 animals-14-03551-f004:**
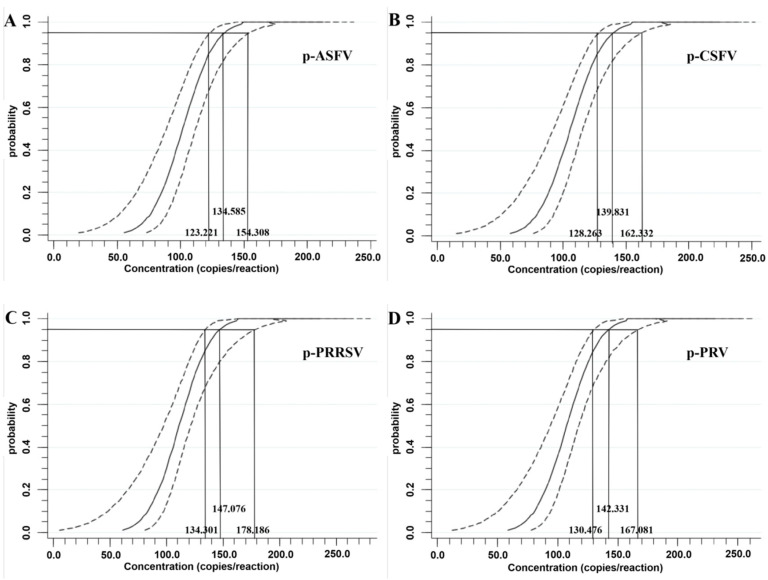
Sensitivity assessment of the quadruplex RT-qPCR using a Probit regression analysis. The LODs of p-ASFV (**A**), p-CSFV (**B**), p-PRRSV (**C**), and p-PRV (**D**) were determined to be 134.585 (95% CI of 123.221–154.308), 139.831 (95% CI of 128.263–162.332), 147.076 (95% CI of 134.301–178.186), and 142.331 (95% CI of 130.476–167.081) copies/reaction, respectively.

**Table 1 animals-14-03551-t001:** The primers and TaqMan probes.

Virus	Gene	Primer/Probe	Sequence (5′ → 3′)	Product (bp)
		ASFV-F ASFV-R ASFV-P	CAAAGTTCTGCAGCTCTTACA	
ASFV	*B646L* (p72)		TGGGTTGGTATTCCTCCCGT	120
			FAM-TCCGGGYGCGATGATGATTACCTT-BHQ1	
		CSFV-F CSFV-R CSFV-P	TAGTGGCGAGCTCCCTGGGTG	
CSFV	*5′UTR*		GGTTAAGGTGTGTCTTGGGCAT	124
			VIC-AGTACAGGACAGTCGTCAGTAGTTCGACGT-BHQ1	
		PRRSV-F PRRSV-R PRRSV-P	TTCATCACYTCCAGATGC	
PRRSV	*ORF6*		GCCRCCCAACACGAGGC	195
			CY5-TACATTCTGGCCCCTGCCCACC-BHQ2	
		PRV-F PRV-R PRV-P	GAGGCCCTGGAAGAAGTT	
PRV	*gB*		TCCTGGACTACAGCGAGAT	131
			Texas red-ATGCCGCGCAGCAGCACCAC-BHQ2	

Note: R = A + G and Y = T + C.

**Table 2 animals-14-03551-t002:** The reaction system of the multiplex RT-qPCR.

Reagent	Volume (μL)	Final Concentration (nM)
2× One-Step RT-PCR Buffer	10.0	/
Ex Taq HS (5 U/μL)	0.4	/
PrimerScript RT Enzyme Mix	0.4	/
ASFV-F (20 pmol/μL)	0.2	200
ASFV-R (20 pmol/μL)	0.2	200
ASFV-P (20 pmol/μL)	0.2	200
CSFV-F (20 pmol/μL)	0.4	400
CSFV-R (20 pmol/μL)	0.4	400
CSFV-P (20 pmol/μL)	0.3	300
PRRSV-F (20 pmol/μL)	0.2	200
PRRSV-R (20 pmol/μL)	0.2	200
PRRSV-P (20 pmol/μL)	0.2	200
PRV-F (20 pmol/μL)	0.4	400
PRV-R (20 pmol/μL)	0.4	400
PRV-P (20 pmol/μL)	0.4	400
Nucleic Acid Template	2.0	/
Nuclease-Free Water	3.5	/

**Table 3 animals-14-03551-t003:** The Ct values and hit rates of the serially diluted plasmids.

Plasmid	Concentration (Copies/Reaction)	Number of Samples	Quadruplex RT-qPCR	
			Ct (Average)	Hit Rate (%)
p-ASFV	500	30	33.47	100
	250	30	34.53	100
	125	30	35.79	90.00
	62.5	30	ND	0
p-CSFV	500	30	33.81	100
	250	30	35.00	100
	125	30	35.92	83.33
	62.5	30	ND	0
p-PRRSV	500	30	33.72	100
	250	30	35.07	100
	125	30	35.90	73.33
	62.5	30	ND	0
p-PRV	500	30	33.59	100
	250	30	34.66	100
	125	30	35.89	80.00
	62.5	30	ND	0

**Table 4 animals-14-03551-t004:** Reproducibility of the multiplex RT-qPCR.

Plasmid	Concentration (Copies/μL)	Intra-Assay Ct Value			Inter-Assay Ct Value		
		$\bar{X}$	SD	CV (%)	$\bar{X}$	SD	CV (%)
p-ASFV	10^7^	15.579	0.038	0.25	15.552	0.147	0.95
	10^5^	21.189	0.03	0.14	21.185	0.187	0.88
	10^3^	28.019	0.074	0.26	28.028	0.278	0.99
p-CSFV	10^7^	15.532	0.048	0.31	15.423	0.156	1.01
	10^5^	22.829	0.013	0.06	22.528	0.253	1.12
	10^3^	28.174	0.13	0.46	28.106	0.247	0.88
p-PRRSV	10^7^	16.15	0.043	0.26	16.204	0.141	0.87
	10^5^	21.444	0.05	0.23	21.259	0.104	0.49
	10^3^	28.541	0.047	0.17	28.579	0.120	0.42
p-PRV	10^7^	15.069	0.056	0.37	15.028	0.098	0.65
	10^5^	21.119	0.013	0.06	21.061	0.214	1.02
	10^3^	27.945	0.042	0.15	27.967	0.247	0.88

**Table 5 animals-14-03551-t005:** Assessment results of clinical specimens.

City	Number	Number of Positive Specimen										
		ASFV	CSFV	PRRSV	PRV	ASFV + CSFV + PRRSV	ASFV+ PRRSV + PRV	ASFV + CSFV	ASFV + PRRSV	ASFV + PRV	PRRSV + PRV	CSFV + PRRSV
Nanning	357	1	2	49	6	0	0	0	1	0	1	0
Guilin	31	0	0	5	1	0	0	0	0	0	0	0
Liuzhou	370	101	8	46	5	0	0	1	6	1	0	4
Beihai	13	9	0	5	0	0	0	0	0	0	0	0
Yulin	561	149	8	107	21	1	2	1	23	3	5	2
Wuzhou	10	0	0	0	0	0	0	0	0	0	0	0
Qinzhou	10	0	1	4	0	0	0	0	0	0	0	0
Baise	588	1	5	40	5	0	0	0	0	0	2	0
Hezhou	72	1	2	23	1	0	0	0	0	0	0	0
Guigang	168	3	0	28	1	0	0	0	0	0	0	0
Hechi	15	0	0	0	0	0	0	0	0	0	0	0
Fangchenggang	347	5	0	36	1	0	0	0	0	0	0	0
Laibin	80	5	0	8	0	0	0	0	0	0	0	0
Chongzuo	494	63	0	114	2	0	0	0	5	0	0	0
Total	3116	338	25	465	43	1	2	2	35	4	8	6
Positivity Rate (%)		10.85%	0.80%	14.92%	1.38%	0.03%	0.06%	0.06%	1.12%	0.13%	0.26%	0.19%

**Table 6 animals-14-03551-t006:** Assessment results of the clinical specimens from different originations.

Source	Number	Number of Positive Specimen										
		ASFV	CSFV	PRRSV	PRV	ASFV + CSFV + PRRSV	ASFV+ PRRSV + PRV	ASFV + CSFV	ASFV + PRRSV	ASFV + PRV	PRRSV + PRV	CSFV + PRRSV
Pig Farm	369	2	6	49	11	0	0	1	4	0	1	1
		(0.54%)	(1.63%)	(13.28%)	(2.98%)			(0.27%)	(1.08%)		(0.27%)	(0.27%)
Slaughterhouse	1784	78	1	229	4	0	0	0	0	0	0	0
		(4.37%)	(0.06%)	(12.84%)	(0.22%)							
Harmless Treatment Plant	963	258	18	187	28	1	2	1	31	4	7	5
		(26.79%)	(1.87%)	(19.42%)	(2.91%)	(0.10%)	(0.21%)	(0.10%)	(3.22%)	(0.42%)	(0.73%)	(0.52%)
Total	3116	338	25	465	43	1	2	2	35	4	8	6
		(10.85%)	(0.80%)	(14.92%)	(1.38%)	(0.03%)	(0.06%)	(0.06%)	(1.12%)	(0.13%)	(0.26%)	(0.19%)

**Table 7 animals-14-03551-t007:** The diagnostic sensitivity and specificity of the established assay.

The Established Quadruplex RT-qPCR		The Reference qPCR		Total	Diagnostic Sensitivity (95% CI)	Diagnostic Specificity (95% CI)
		Positive	Negative			
ASFV	Positive	330	8	338	99.10% (97.39–99.69%)	99.71% (99.43–99.85%)
	Negative	3	2775	2778		
	Total	333	2783	3116		
CSFV	Positive	23	2	25	95.83% (79.76–99.26%)	99.94% (99.76–99.98%)
	Negative	1	3090	3091		
	Total	24	3092	3116		
PRRSV	Positive	452	13	465	99.12% (97.77–99.66%)	99.51% (99.17–99.71%)
	Negative	4	2647	2651		
	Total	456	2660	3116		
PRV	Positive	39	4	43	97.50% (87.12–99.56%)	99.87% (99.67–99.95%)
	Negative	1	3072	3073		
	Total	40	3076	3116		

**Table 8 animals-14-03551-t008:** Coincidence rate between the established and reference assays.

Method	Number of Positive Sample			
	ASFV	CSFV	PRRSV	PRV
The developed assay	338/3116 (10.84%)	25/3116 (0.80%)	465/3116 (14.92%)	43/3116 (1.38%)
The reference assay	333/3116 (10.69%)	24/3116 (0.77%)	456/3116 (14.63%)	40/3116 (1.28%)
Coincidence rate	99.65%	99.90%	99.45%	99.84%

## Data Availability

The original contributions presented in the study are included in the article/Appendix A, further inquiries can be directed to the corresponding authors.

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
