# Peer review of "A Quadruplex RT-qPCR for the Detection of African Swine Fever Virus, Classical Swine Fever Virus, Porcine Reproductive and Respiratory Syndrome Virus, and Porcine Pseudorabies Virus"

_animals, 2024, doi:10.3390/ani14233551_

Round 1

Reviewer 1 Report

Comments and Suggestions for Authors

MAJOR COMMENTS

The study describes the validation of a quadruplex real time PCR for ASF, CSF, PRV and PRRS. Although interesting and useful for the local context, the study needs some improvements.

The methodology is not always detailed and no description on test optimization is provided. In addition, there is no reference to reference tests as a singlepex test in the introduction (i.e. the necessity to further develop and additional real time for ASFV when many are available).

Materials and method sections needs improvement adding details and information at the moment missing. The result section is difficult to follow as many data are presented in too many tables not easy to interpret.

The discussion section does not take into account all data presented and should be re-written.

INTRODUCTION

The introduction provides general information to the reader to understand the study performed, however it seems the choice of the four pathogens to be targeted to the Chinese context. A tetraplex real Time PCR is not always affordable for every labs and the viruses identified may vary in distribution and prevalence according to the regional context.

Regarding the similarity of clinical signs between ASF and CSF, this is agreeable but with the other viruses the statement should be refined.

Please add more information on previous studies on the development of duplex and/or triplex and/or quadruplex real time assay, in order to let the reader understand the necessity to develop and additional real time PCR assay.

MATERIAL AND METHODS

Paragraph 2.1: please dived the list of viruses used for the development of the assay from all the others used for the specificity. It could help the reader to quantify the number of reference strains used for the development of the assay and for specificity.

If possible, include the genbank accession number for each strain used.

Please specify the SIV subtype used and in particular, the ASFV strains used: name and accession numbers. This information render the study reproducible by other labs.

Please in table 1 add the reference is available of primers and probe used. Reading the paper it seem that for ASFV it was used a set of primers and probes already available, if not please clarify

There are no description of the singlepex used as reference method, anyway showed in table 8. In addition, add information on the sensitivity and specificity of each reference method used to evaluate data presented.

Looking at the data in the supplementary material and in particular to the alignment used to design primers, it appears the necessity to use some degenerations not present in the primer sequence. Please provide explanation for that.

All tests to optimize the quadruplex are not described, these are the core of the study.

Please add how the annealing temperature was identified.

Paragraph 2.8: the target viruses were used for inclusivity and the other for strict specificity.

Paragraph 2.10 how many operators were used?

Paragraph 2.11 please provide details on which WOAH test was used for ASFV as two distinct real time PCR are described. In addition this information is in contrast with the information provided in the introduction lines 108-110.

Paragraph 2.9 Please provide additional description on how the LOD was calculated: i.e. number of replicates? If tested in one replicate please confirm with more replicates.

RESULT

No data are provided on the optimization phase of the study. Please provide full criteria for positive and negative results and how they were selected.

No internal controls were foresaw in the assay, provide justification for that.

In figure 2 plots of the specificity test are reported. In each plot (4 for the 4 target virus) it is always present a fluorescence signal for the other 3 please provide an explanation and details for that.

In figure 2A (signals for ASFV), it is not present any signal for the sample number 7 that should be ASF why?

All figures need a better description.

Table 3 why 30 samples? Please add this info in the materials and methods section.

The diagnostic sensitivity should be compared with a singlex validated assay or anyway a test considered a gold standard in order to evaluate the results. This is presented in Table 8 but not in the materials and methods section where other reference methods only for ASFV and CSF are reported (i.e. WOAH manual)

Table 7 is difficult to read and interpret.

There are too many tables difficult to follow.

DISCUSSION

Please provide information on why a set of primers and probes were newly designed for ASF and CSF when several are already available starting from the WOAH Manual.

The discussion should focus more on the developed test rather than on the epidemiology and prevalence of the four viruses tested.

No comments are provided regarding samples not detected with the developed assay in comparison with the reference standard.

Discussion should be thoroughly revised in order to comment the performances of the developed method in relation to the reference methods.

MINOR COMMENTS

ABSTRACT

Line 33: Please modify “for all other swine viruses” as only a selection of swine viruses were tested for specificity and not all.

INTRODUCTION

ASF was described in 1921, but detect in 1910 in Africa by Montgomery (1921), please modify.

Line 53: please substitute with 3rd August 2018

Line 73: please use eradicated instead of eliminated

MATERIAL AND METHODS

Please in paragraph 2.4 use g instead of rpm

Add information on the setting of the homogeniser (herts/min).

Please be consistent when using copies per reaction o microliters as it may generate confusion.

Figure 1: please in the notes specify what the number 1-8 stand for.

Figure 2 please provide notes for all the numbers present in all plots. It is really confusing.

Author Response

The Cover Letter

November 26, 2024

Dear editor,

The manuscript has been revised carefully according to the suggestions of the reviewers and editor. The details are as follows.

Reviewer #1

MAJOR COMMENTS

The study describes the validation of a quadruplex real time PCR for ASF, CSF, PRV and PRRS. Although interesting and useful for the local context, the study needs some improvements.

  1. The methodology is not always detailed and no description on test optimization is provided.

Response: We agree to the reviewer's suggestion. The methodology and description on test optimization have been described in the Section 2.6 in the revised manuscript. Please Lines 203-219 in the revised manuscript.

  1. In addition, there is no reference to reference tests as a singlepex test in the introduction (i.e. the necessity to further develop and additional real time for ASFV when many are available).

Response: We agree to the reviewer's suggestion. The necessity to further develop an additional quadruplex RT-qPCR for the detection of ASFV, CSFV, PRRSV, and PRV has been added in the Introduction section in the revised manuscript. Please see Lines 115-126 in the revised manuscript.

  1. Materials and method sections needs improvement adding details and information at the moment missing.

Response: We agree to the reviewer's suggestion. The Materials and Methods section has been revised carefully according to the reviewers’ suggestion. The details can be seen in the following responses to the reviewers. Please see the revised manuscript.

  1. The result section is difficult to follow as many data are presented in too many tables not easy to interpret.

Response: Each table in this manuscript provide different information. Table 1 provides the information on the specific primers and probes. Table 2 provides the reaction system of the multiplex RT-qPCR. Table 3 shows the data information related to Probit regression analysis. Table 4 indicates the reproducibility data. Table 5 represents the detection results of the 3116 clinical samples using the developed assay. Table 6 represents the detection results of the clinical samples originated from different sources. Table 7 shows the diagnostic sensitivity and specificity of the established assay. Table 8 represents the coincidence rates of the developed and the reference assays. They are all necessary. Please see Table 1-7 in the revised manuscript.

  1. The discussion section does not take into account all data presented and should be re-written.

Response: We agree to the reviewer's suggestion. The Discussion section has been revised carefully. Please see the revised manuscript.

INTRODUCTION

  1. The introduction provides general information to the reader to understand the study performed, however it seems the choice of the four pathogens to be targeted to the Chinese context. A tetraplex real Time PCR is not always affordable for every lab and the viruses identified may vary in distribution and prevalence according to the regional context.

Response: We agree to the reviewer’s suggestion. At present, ASFV, CSFV, PRRSV, and PRV are still circulating in pig herds in China and other countries. Therefore, the genomic sequences of the representative strains from different countries and dates were downloaded from NCBI GenBank, and the multiple sequence alignments were performed. Then, the specific primers and probes were designed targeting the conserved regions of the B646L (p72) gene of ASFV, the 5′ untranslated region (5'UTR) of CSFV, the ORF6 gene of PRRSV, and the gB gene of PRV. After optimizing the reaction conditions, a quadruplex RT-qPCR assay was developed for the differential detection of ASFV, CSFV, PRRSV, and PRV. Therefore, the developed quadruplex RT-qPCR not only can detect different viral genotypes and strains from China, but also can detect different viral genotypes and strains from other countries. Please see Lines 168-179 and Table 1 in the revised manuscript.

  1. Regarding the similarity of clinical signs between ASF and CSF, this is agreeable but with the other viruses the statement should be refined.

Response: We agree to the reviewer’s suggestion. Pigs infected with ASFV, CSFV, PRRSV, and/or PRV show similar signs such as increased body temperature and loss of appetite, and are prone to respiratory symptoms, digestive symptoms, neurological symptoms, and/or abortion in pregnant sows.

Since ASFV, CSFV, PRRSV, and PRV are still circulating in pig herds in China and other countries, and they cause similar clinical manifestations and pathological damages, which challenging the accuracy of differential diagnosis of them through only clinical signs and gross pathological changes. Therefore, a one-step multiplex TaqMan RT-qPCR assay is developed for the rapid, and specific differential detection of ASFV, PRRSV, CSFV, and PRV in this study. Please see Lines 50-53, 108-112 and 124-126 in the revised manuscript.

  1. Please add more information on previous studies on the development of duplex and/or triplex and/or quadruplex real time assay, in order to let the reader understand the necessity to develop an additional real time PCR assay.

Response: We agree to the reviewer's suggestion. More information on previous studies on the development of duplex and/or triplex and/or quadruplex real time assay has been added in the revised manuscript. Please see Lines 115-126 in the revised manuscript.

MATERIAL AND METHODS

  1. Paragraph 2.1: please dived the list of viruses used for the development of the assay from all the others used for the specificity. It could help the reader to quantify the number of reference strains used for the development of the assay and for specificity.

Response: We agree to the reviewer's suggestion. The vaccine strains and positive samples used to construct the recombinant standard plasmid constructs, and used as the positive controls in the developed assay’s specificity analysis, and those used as controls in the developed assay’s specificity analysis were separately described in the revised manuscript. Please see Lines 135-159 in the revised manuscript.

  1. If possible, include the genbank accession number for each strain used.

Response: We agree to the reviewer's suggestion. The viral strains used in this study included vaccine strains for construction of standard constructs, and the reference strains for design of specific primers and probes. The GenBank accession numbers of these strains have been listed in the Supplementary Table S1-S4. Due to limitations in the length of the manuscript, these accession numbers are not listed in the main text. Please see Lines 158-159 and Supplementary Table S1-S4 in the revised manuscript.

  1. Please specify the SIV subtype used and in particular, the ASFV strains used: name and accession numbers. This information renders the study reproducible by other labs.

Response: We agree to the reviewer's suggestion. The H1N1 subtype SIV has been added in the revised manuscript. Please see Line 143 in the revised manuscript.

The clinical positive samples of genotype I ASFV and genotype II ASFV were provided by our laboratory. They have been confirmed by PCR and gene sequence analysis, but did not upload to GenBank. In addition, the information on the ASFV strains used for primer and probe design is described in the Supplementary Table S1. Please see Supplementary Table S1 in the revised manuscript.

  1. Please in table 1 add the reference is available of primers and probe used. Reading the paper it seem that for ASFV it was used a set of primers and probes already available, if not please clarify.

Response: We agree to the reviewer's suggestion. The specific primers and probe for detection of ASFV in Table 1 were derived from reference 8. The reference 8 is mentioned in the revised manuscript. Please see Lines 175-178 and the reference 8 in the revised manuscript.

  1. There are no description of the singlepex used as reference method, anyway showed in table 8. In addition, add information on the sensitivity and specificity of each reference method used to evaluate data presented.

Response: We agree to the reviewer's suggestion. Our manuscript focused on the development of quadruplex RT-qPCR for the detection of ASFV, CSFV, PRRSV, and PRV. The reference singleplex RT-qPCR assays in this study were used to test the clinical samples, and the results were compared with the results of the developed assay. Therefore, these reference assays were only mentioned in shortly. Please see Lines 255-264 in the revised manuscript.

In addition, the sensitivity and specificity of each reference method were discussed in the Section Discussion. Please see Lines 425-427 in the revised manuscript.

  1. Looking at the data in the supplementary material and in particular to the alignment used to design primers, it appears the necessity to use some degenerations not present in the primer sequence. Please provide explanation for that.

Response: We agree to the reviewer's suggestion. In this study, the specific primers and probes are designed to avoid the use of degenerate bases as much as possible. The degenerate bases lead to negative effects such as reduced specificity, and amplification efficiency. Mutations at a single site in individual strains have a small effect on sequence amplification. Therefore, we did not use degenerate bases in this study. Please see Table 1 and Supplementary Figure S1 in the revised manuscript.

According to the results in this study, the designed specific primers and probes can sensitively and specifically detect the clinical samples. Please see the revised manuscript.

  1. All tests to optimize the quadruplex are not described, these are the core of the study.

Response: We agree to the reviewer's suggestion. The details to optimize the quadruplex RT-qPCR have been described in the revised manuscript. Please see Lines 203-219 in the revised manuscript.

  1. Please add how the annealing temperature was identified.

Response: We agree to the reviewer's suggestion. In this study, different annealing temperatures (54, 55, 56, 57, 58, 59, 60, 61and 62 ℃) were used for amplification to obtain the optimal annealing temperature. The optimal conditions were determined based on the minimum cycling threshold (Ct) and the maximum ∆Rn. Finally, the optimal annealing temperature was 55℃. Please see Paragraph 2.6 and 3.2 in the revised manuscript.

  1. Paragraph 2.8: the target viruses were used for inclusivity and the other for strict specificity.

Response: We agree to the reviewer's suggestion. The target viruses were used for inclusivity and the other for strict specificity. This content has been added in the revised manuscript. Please see Lines 225-233 in the revised manuscript.

  1. Paragraph 2.10 how many operators were used?

Response: In this study, the reproducibility analysis was done by one operator. The experiments were to analyze the stability and reliability of the developed quadruplex RT-qPCR. The experiment repeated three times, and performed at three different days. Please see Lines 244-249 of Paragraph 2.10 in the revised manuscript.

  1. Paragraph 2.11 please provide details on which WOAH test was used for ASFV as two distinct real time PCR are described. In addition, this information is in contrast with the information provided in the introduction lines 108-110.

Response: We agree to the reviewer's suggestion. The qPCR/RT-qPCR outlined in the World Organisation for Animal Health (WOAH)'s Terrestrial Manuals for ASFV, CSFV, and PRV, and the RT-qPCR outlined in the Chinese entry-exit inspection and quarantine industry standard for PRRSV were used as reference methods to detect the clinical samples. In this study, it was focused on the development of the quadruplex RT-qPCR for the detection of ASFV, CSFV, PRRSV, and PRV. Therefore, the reference assays were mentioned, and the websites of these assays were provided. The details of these methods can be consulted in these Terrestrial Manuals. Please see Lines 255-264 in the revised manuscript.

In the Introduction section, it was only described that the singleplex qPCR/RT-qPCR assays for the detection of ASFV, CSFV, PRRSV, or PRV have been reported, and these assays used one pair of specific primers and one probe to detect one virus of ASFV, CSFV, PRRSV, or PRV. However, these assays did not be used as reference assays to detect these viruses. The assays recommended by WOAH and China Entry-Exit Inspection and Quarantine Bureau were used as references in this study. Please see Lines 115-118 and 255-264 in the revised manuscript.

  1. Paragraph 2.9 Please provide additional description on how the LOD was calculated: i.e. number of replicates? If tested in one replicate please confirm with more replicates.

Response: We agree to the reviewer's suggestion. In this study, the LODs were determined using two methods, 10-fold serially dilution method and Probit regression analysis. The fluorescence amplification plot showed that the lowest detection concentration of the assay was 101 copies/μL (200 copies/reaction) after ten-fold serially dilution. In addition, Probit regression analysis was used to calculate LOD. The results showed that the LODs were 134.585, 139.831, 147.076, and 142.331 copies/reaction for ASFV, CSFV, PRRSV, and PRV, respectively. Two methods showed similar results. Please see Lines 317-328, Figure 3 and Figure 4 in the revised manuscript.

RESULT

  1. No data are provided on the optimization phase of the study. Please provide full criteria for positive and negative results and how they were selected.

Response: We agree to the reviewer's suggestion. To optimize the reaction conditions, different concentrations of primers and probes, different annealing temperatures and different reaction cycles were combined to perform tests. Finally, the optimal concentrations, temperatures and cycles were determined. Our manuscript only provides the optimal reaction conditions. In addition, the criteria for positive and negative results have been added in the revised manuscript. Please see Lines 275-286 and Table 2 in the revised manuscript.

  1. No internal controls were foresaw in the assay, provide justification for that.

Response: We agree to the reviewer's suggestion. In this study, the recombinant plasmid constructs were generated to used as the standard plasmid constructs. They were used to develop the quadruplex RT-qPCR, and used as positive control in the process of experiments. In addition, the clinical positive samples were used as positive controls, and the clinical negative samples and nuclease-free distilled water were used as negative samples. Please see Lines 191-201 and Lines 225-233 in the revised manuscript.

  1. In figure 2 plots of the specificity test are reported. In each plot (4 for the 4 target virus) it is always present a fluorescence signal for the other 3. Please provide an explanation and details for that.

Response: We agree to the reviewer's suggestion. In each test for specificity analysis, the mixture of plasmid constructs p-ASFV, p-CSFV, p-PRRSV, and p-PRV was used as positive control. Therefore, the amplification curves of p-ASFV, p-CSFV, p-PRRSV, and p-PRV were shown in each plot. Please see Figure 2 in the revised manuscript.

  1. In figure 2A (signals for ASFV), it is not present any signal for the sample number 7 that should be ASF. why?

Response: We agree to the reviewer's suggestion. The No. 7 sample is the positive clinical sample of CSFV, so it did not show any fluorescence signal when the specific primer and probe for ASFV were used. Please see Lines 309 in the revised manuscript.

  1. All figures need a better description.

Response: We agree to the reviewer's suggestion. All Figures have been checked, and revised carefully to improve the description. Please see Figure 1 to Figure 4 in the revised manuscript.

  1. Table 3 why 30 samples? Please add this info in the materials and methods section.

Response: We agree to the reviewer's suggestion. According to the principle of Probit regression analysis, the sample size should be selected basing on the experimental design and statistical requirements. The aim of this study is to analyze the sensitivity of Quadruplex RT-qPCR with relative accuracy. In clinical trials, many factors can cause experimental errors, such as different personnel, different instruments, etc. It is hard to avoid experimental errors, so the Probit regression analysis should also be allowed to have small errors, and the sample size of 30 has reached the requirement of experimental feasibility. Therefore, 30 samples were used in this study. This information has been added in the revised manuscript. Please see Lines 238-243 in the Materials and Methods section in the revised manuscript.

  1. The diagnostic sensitivity should be compared with a singlex validated assay or anyway a test considered a gold standard in order to evaluate the results. This is presented in Table 8 but not in the materials and methods section where other reference methods only for ASFV and CSF are reported (i.e. WOAH manual).

Response: We agree to the reviewer's suggestion. The diagnostic sensitivity and specificity have been described in the revised manuscript. Please see Lines 265-267 in the Materials and Methods section in the revised manuscript.

The results of diagnostic sensitivity and specificity, and the coincidence rates were described in Section 3.7, Table 7 and Table 8. Please see Lines 370-377, Table 7 and Table 8 in the revised manuscript.

  1. Table 7 is difficult to read and interpret.

Response: We agree to the reviewer's suggestion. The detection results of the clinical samples using the developed assay and the reference assays are shown in Table 7. In addition, the results of these two assays are compared, and the diagnostic sensitivity and specificity are calculated and shown in Table 7.  Please see Table 7 in the revised manuscript.

  1. There are too many tables difficult to follow.

Response: Each table in this manuscript provide different information. Table 1 provides the information on the specific primers and probes. Table 2 provides the reaction system of the multiplex RT-qPCR. Table 3 shows the data information related to Probit regression analysis. Table 4 indicates the reproducibility data. Table 5 represents the detection results of the 3116 clinical samples using the developed assay. Table 6 represents the detection results of the clinical samples originated from different sources. Table 7 shows the clinical sensitivity and specificity of the established assay. Table 8 represents the coincidence rates of the developed and the reference assays. They are all necessary. Please see Table 1-7 in the revised manuscript.

DISCUSSION

  1. Please provide information on why a set of primers and probes were newly designed for ASF and CSF when several are already available starting from the WOAH Manual.

Response: We agree to the reviewer's suggestion. The primers and probes used in this study are all designed in our laboratory. Those of ASFV and CSFV have been reported in other papers (reference 8 and 39), and those of PRRSV and PRV are first used in this manuscript. All the primers and probes used in the developed quadruplex RT-qPCR assay come from our laboratory. The reference assays come from the WOAH Manuals and Chinese entry-exit inspection and quarantine industry standard, and provide corresponding primers and probes. The information has been added in the Discussion section in the revised manuscript. Please see Lines 418-421 in the revised manuscript.

  1. The discussion should focus more on the developed test rather than on the epidemiology and prevalence of the four viruses tested.

Response: We agree to the reviewer's suggestion. The discussion has focused on the developed test, and some contents of the epidemiology and prevalence of the four viruses have been deleted. Please see the Discussion section in the revised manuscript.

  1. No comments are provided regarding samples not detected with the developed assay in comparison with the reference standard.

Response: We agree to the reviewer's suggestion. The comments are provided regarding samples not detected with the developed assay in comparison with the reference standard. Please see Lines 33-437 in the revised manuscript.

  1. Discussion should be thoroughly revised in order to comment the performances of the developed method in relation to the reference methods.

Response: We agree to the reviewer's suggestion. The Discussion section has been thoroughly revised. Please see the revised manuscript.

MINOR COMMENTS

ABSTRACT

  1. Line 33: Please modify “for all other swine viruses” as only a selection of swine viruses were tested for specificity and not all.

Response: We agree to the reviewer's suggestion. Revise “for all other swine viruses” to “for the other control swine viruses used in this study”. Please see Line 35 in the revised manuscript.

INTRODUCTION

  1. ASF was described in 1921, but detect in 1910 in Africa by Montgomery (1921), please modify.

Response: We agree to the reviewer's suggestion. This has been revised from “found” to “described”. Please see Line 55 in the revised manuscript.

  1. Line 53: please substitute with 3rdAugust 2018.

Response: We agree to the reviewer's suggestion. Revise “3 August 2018” to “3rd August 2018”. Please see Line 57 in the revised manuscript.

  1. Line 73: please use eradicated instead of eliminated.

Response: We agree to the reviewer's suggestion. Revise “eliminated” to “eradicated”.  Please see Line 78-79 in the revised manuscript.

MATERIAL AND METHODS

  1. Please in paragraph 2.4 use ginstead of rpm.

Response: We agree to the reviewer's suggestion. Revise “rpm” to “g”.  Please see Line 187 in the revised manuscript.

  1. Add information on the setting of the homogeniser (herts/min).

Response: We agree to the reviewer's suggestion. The information on the setting of the homogeniser has been added. Please see Line 187 in the revised manuscript.

  1. Please be consistent when using copies per reaction or microliters as it may generate confusion.

Response: We agree to the reviewer's suggestion. The copies/μL was used in this study. In addition, while the assay’s sensitivity was analyzed, the copies/reaction was used. The LODs were determined using two methods, 10-fold serially dilution method and Probit regression analysis. The fluorescence amplification plot showed that the lowest detection concentration of the assay was 101 copies/μL (200 copies/reaction) after ten-fold serially dilution. In addition, Probit regression analysis was used to calculate LOD. The results showed that the LODs were 134.585, 139.831, 147.076, and 142.331 copies/reaction for ASFV, CSFV, PRRSV, and PRV, respectively. Two methods showed similar results. Please see Lines 317-326, Figure 3 and Figure 4 in the revised manuscript.

  1. Figure 1: please in the notes specify what the number 1-8 stand for.

Response: We agree to the reviewer's suggestion. The meanings of the number 1-8 in the Figure 1 have been added in the revised manuscript. Please see Lines 296-298 in the revised manuscript.

  1. Figure 2: please provide notes for all the numbers present in all plots. It is really confusing.

Response: We agree to the reviewer's suggestion. The notes have been provided for all the numbers in the Figure 2. Please see Lines 307-316 and Figure 2 in the revised manuscript.

Reviewer 2 Report

Comments and Suggestions for Authors

Comments

Abstract

Lane 26, change the word “Symptoms” to “clinical signs”, this is requested throughout the document.

Lane 36, please indicate the year in which the clinical samples analyzed were obtained.

Introduction

Indicate what would be the advantages of doing a multiplex test that identifies ASFV, CSFV, PRRSV and PRV.

Describe what possible disadvantages or advantages there are in including in a multiplex test an RNA virus and three DNA viruses.

M&M

Reference strains, in the case of strains provided by your own laboratory, do you not have any reference, for example, a GenBank accession number or other? This would be equally useful for strains obtained from other sources.

Indicate how the organ macerates were prepared.

Lane 144, indicate in the text how many sequences you included in the analysis for each virus.

Lane 172-180, How much DNA/RNA did you use in the reaction? What was the average concentration of the samples analyzed?

Results

Figure 2. The figure is very difficult to understand, can you separate the graphs for a better understanding? This can be done in the supplementary material.

When constructing the calibration curves you obtain the absolute quantification value, why did you not record it for the samples, can you present an average with SD

Author Response

The Cover Letter

November 26, 2024

Dear editor,

The manuscript has been revised carefully according to the suggestions of the reviewers and editor. The details are as follows.

Reviewer #2

Comments and Suggestions for Authors

Comments

 Abstract

  1. Lane 26, change the word “Symptoms” to “clinical signs”, this is requested throughout the document.

Response: We agree to the reviewer's suggestion. Revise “Symptoms” to “clinical signs” throughout the document. Please see the revised manuscript.

  1. Lane 36, please indicate the year in which the clinical samples analyzed were obtained.

 Response: We agree to the reviewer's suggestion. The collected date of the clinical samples has been added in the revised manuscript. Please see Lines 38-39 in the revised manuscript.

Introduction

  1. Indicate what would be the advantages of doing a multiplex test that identifies ASFV, CSFV, PRRSV and PRV.

Response: We agree to the reviewer's suggestion. The advantages of doing a multiplex test that identifies ASFV, CSFV, PRRSV and PRV have been added in the revised manuscript. Please see Lines 126-129 in the revised manuscript.

  1. Describe what possible disadvantages or advantages there are in including in a multiplex test an RNA virus and three DNA viruses.

Response: We agree to the reviewer's suggestion. The possible disadvantages or advantages of the developed assay have been described in the revised manuscript. Please see Lines 129-132 in the revised manuscript.

M&M

  1. Reference strains, in the case of strains provided by your own laboratory, do you not have any reference, for example, a GenBank accession number or other? This would be equally useful for strains obtained from other sources.

Response: We agree to the reviewer's suggestion. The reference strains that can be obtained their GenBank accession numbers have been described in the Supplementary Table S1-S4. The clinical positive samples were provided by our laboratory. They have been confirmed by PCR and gene sequence analysis, but did not upload to GenBank.

  1. Indicate how the organ macerates were prepared.

Response: We agree to the reviewer's suggestion. The details to prepare the tissues have been added in the revised manuscript. Please see Lines 182-190 in the revised manuscript.

  1. Lane 144, indicate in the text how many sequences you included in the analysis for each virus.

Response: We agree to the reviewer's suggestion. The numbers of sequences for each virus have been added in the revised manuscript. Please see Lines 169-170 in the revised manuscript.

  1. Lane 172-180, How much DNA/RNA did you use in the reaction? What was the average concentration of the samples analyzed?

Response: The recombinant plasmid constructs (p-ASFV, p-CSFV, p-PRRSV, and p-PRV) were used as standard plasmid constructs, and used to optimize the optimal reaction conditions. The mixture of the four plasmid constructs with concentrations of 107 copies/μL was used to optimize the annealing temperatures and reaction cycles. After optimization of the annealing temperatures and reaction cycles, different concentrations of the plasmid constructs were used to perform the experiments, and the optimal concentrations were finally determined. Please see Lines 203-219 in the revised manuscript.

Results

  1. Figure 2. The figure is very difficult to understand, can you separate the graphs for a better understanding? This can be done in the supplementary material.

Response: We agree to the reviewer's suggestion. The notes of Figure 2 have been redescribed to be understood better in the revised manuscript. Please see Lines 307-316 and Figure 2 in the revised manuscript.

  1. When constructing the calibration curves you obtain the absolute quantification value, why did you not record it for the samples, can you present an average with SD.

Response: The RT-qPCR can obtain the absolute quantification value in the samples. In this study, the purpose of detecting the clinical samples was determined whether the samples were positive samples or not. Therefore, the positive clinical samples were recorded, and the average quantities with SD of the samples were not recorded. Please see Lines 349-366 and Table 5 in the revised manuscript.

Reviewer 3 Report

Comments and Suggestions for Authors

The presented work is devoted to the development, validation and verification on a large number of samples (3116) of a quadruplex PCR assay for the simultaneous detection of four important viruses of pigs: ASFV, CSFV, PPRV and PRV. The scientific novelty of the work is not very high, since, as indicated by the authors, PCR tests for detection of these viruses separately or in multiplex format already exists. At the same time, the practical significance of the work is very high, since the use of the PCR test developed by the authors will reduce the time for monitoring these particularly dangerous pig diseases. The manuscript is of practical interest to scientists involved in the diagnosis and monitoring of pig diseases. It is necessary to note some controversial points in the work:

In the introduction, the authors state that it is impossible to differentiate these 4 diseases by pathomorphological signs. One can agree that ASF and CSF are indeed impossible to distinguish. As for PPRV and PRV, the complex of symptoms will depend on the form of the disease, the age of the animals and a number of other factors. For example, PPRV can be asymptomatic in adult animals, but cause abortion in sows and the death of newborn piglets. However, it should be recognized that a similar complex of symptoms can be observed with each of the four diseases, but not always.

One of the possible weak points of the work, as I see it, is the determination of the sensitivity of the test using a series of tenfold dilutions of plasmids carrying the target insertion. The fact is that this method does not allow taking into account the effectiveness of reverse transcription, since DNA plasmids are used to assess sensitivity, and CSFV and PPRV are RNA-containing viruses. More accurate results could be obtained by studying a series of tenfold dilutions of the CSFV and PPRV samples with a known titer expressed in decimal logarithms. In practice, the assessment of the test sensitivity by this method shows somewhat weaker results than studying a series of plasmid dilutions.

Figure 2 looks quite difficult to read. Perhaps it is not worth overloading the graph with controls for each of the 4 viruses, but simply making separate graphs for each of them. In addition, the fluorescence curves for PPRV look very weak against the background of the other three viruses. This is probably due to the fluorescence level of the selected fluorophore.

The manuscript is written in understandable English, but the text contains a number of inaccuracies and unsuccessful formulations, for example:

- line 68. The word “discovered” is not appropriate in this context. It is better to use the word “introduced” or “identified”.

- line 143. 2.3. Primers and probes. Please indicate which software was used to analyze the sequences for conservation and design of primers. Also, please indicate which commercial company synthesized the primers and probes for your work.
- line 162. Please, remove “of”.

- sentence from line 344 is a repetition of a sentence from the line 324. Please, remove one of them.

- line 350. “… diagnosis depends on laboratory detection and diagnosis”. Please, rephrase this sentence.

- line 390. It is better not to use word “we”.

The manuscript may be recommended for publication after minor revisions.

Author Response

The Cover Letter

November 26, 2024

Dear editor,

The manuscript has been revised carefully according to the suggestions of the reviewers and editor. The details are as follows.

Reviewer #3

The presented work is devoted to the development, validation and verification on a large number of samples (3116) of a quadruplex PCR assay for the simultaneous detection of four important viruses of pigs: ASFV, CSFV, PRRSV and PRV. The scientific novelty of the work is not very high, since, as indicated by the authors, PCR tests for detection of these viruses separately or in multiplex format already exists. At the same time, the practical significance of the work is very high, since the use of the PCR test developed by the authors will reduce the time for monitoring these particularly dangerous pig diseases. The manuscript is of practical interest to scientists involved in the diagnosis and monitoring of pig diseases. It is necessary to note some controversial points in the work:

  1. In the introduction, the authors state that it is impossible to differentiate these 4 diseases by pathomorphological signs. One can agree that ASF and CSF are indeed impossible to distinguish. As for PPRSV and PRV, the complex of symptoms will depend on the form of the disease, the age of the animals and a number of other factors. For example, PPRSV can be asymptomatic in adult animals, but cause abortion in sows and the death of newborn piglets. However, it should be recognized that a similar complex of symptoms can be observed with each of the four diseases, but not always.

Response: We agree to the reviewer's suggestion. Pigs infected with ASFV, CSFV, PRRSV, and/or PRV show similar signs such as increased body temperature and loss of appetite, and are prone to respiratory symptoms, digestive symptoms, neurological symptoms, and/or abortion in pregnant sows. Surely, just like the reviewer’s suggestion, it should be recognized that a similar complex of symptoms can be observed with each of the four diseases, but not always.

Since ASFV, CSFV, PRRSV, and PRV are still circulating in pig herds in China and other countries, and they cause similar clinical manifestations and pathological damages, which challenging the accuracy of differential diagnosis of them through only clinical signs and gross pathological changes. Therefore, a one-step multiplex TaqMan RT-qPCR assay is developed for the rapid, and specific differential detection of ASFV, PRRSV, CSFV, and PRV in this study. Please see Lines 50-53, 108-112 and 124-126 in the revised manuscript.

  1. One of the possible weak points of the work, as I see it, is the determination of the sensitivity of the test using a series of tenfold dilutions of plasmids carrying the target insertion. The fact is that this method does not allow taking into account the effectiveness of reverse transcription, since DNA plasmids are used to assess sensitivity, and CSFV and PRRSV are RNA-containing viruses. More accurate results could be obtained by studying a series of tenfold dilutions of the CSFV and PPRV samples with a known titer expressed in decimal logarithms. In practice, the assessment of the test sensitivity by this method shows somewhat weaker results than studying a series of plasmid dilutions.

Response: We agree to the reviewer's suggestion. The recombinant plasmid constructs were used to analyze the sensitivity of the developed assay. In this study, the LODs were determined using two methods, 10-fold serially dilution method and Probit regression analysis. The fluorescence amplification plot showed that the lowest detection concentration of the assay was 101 copies/μL (200 copies/reaction) after ten-fold serially dilution. In addition, Probit regression analysis was used to calculate LOD. The results showed that the LODs were 134.585, 139.831, 147.076, and 142.331 copies/reaction for ASFV, CSFV, PRRSV, and PRV, respectively. Two methods showed similar results. Please see Lines 317-326, Figure 3 and Figure 4 in the revised manuscript.

  1. Figure 2 looks quite difficult to read. Perhaps it is not worth overloading the graph with controls for each of the 4 viruses, but simply making separate graphs for each of them. In addition, the fluorescence curves for PRRSV look very weak against the background of the other three viruses. This is probably due to the fluorescence level of the selected fluorophore.

Response: We agree to the reviewer's suggestion. In each test for specificity analysis, the mixture of plasmid constructs p-ASFV, p-CSFV, p-PRRSV, and p-PRV was used as positive control. Therefore, the amplification curves of p-ASFV, p-CSFV, p-PRRSV, and p-PRV were shown in each plot. Please see the Figure 2 in the revised manuscript.

In the Quadruplex RT-qPCR assay, the four fluorophores interact with each other, and no matter how the fluorophores were selected, one curve is always relatively weak. In this study, the fluorescence curve of PRRSV is relatively weak, but its ∆Rn can reach to 500,000, and it did not affect the judgement of the detection results. The amplification curves of PRRSV are shown in Figure 1 and Figure 3. After optimization of reaction conditions, the fluorophore interaction did not affect the experimental results. Please see Figure 2 in the revised manuscript.

The manuscript is written in understandable English, but the text contains a number of inaccuracies and unsuccessful formulations, for example:

  1. - line 68. The word “discovered” is not appropriate in this context. It is better to use the word “introduced” or “identified”.

Response: We agree to the reviewer's suggestion. Revise “discovered” to “identified”.  Please see Line 72 in the revised manuscript.

  1. - line 143. 2.3. Primers and probes. Please indicate which software was used to analyze the sequences for conservation and design of primers. Also, please indicate which commercial company synthesized the primers and probes for your work.

Response: We agree to the reviewer's suggestion. The information has been added in the revised manuscript. Please see Lines 168179 in the revised manuscript.

  1. - line 162. Please, remove “of”.

Response: We agree to the reviewer's suggestion. “of” has been removed. Please see Line 193 in the revised manuscript.

  1. - sentence from line 344 is a repetition of a sentence from the line 324. Please, remove one of them.

Response:  We agree to the reviewer's suggestion. The sentence in previous Line 344 has been deleted. Please see Line 407-408 in the revised manuscript.

  1. - line 350. “… diagnosis depends on laboratory detection and diagnosis”. Please, rephrase this sentence.

Response: We agree to the reviewer's suggestion. The sentence has been rewritten. Please see Line 413-414 in the revised manuscript.

  1. - line 390. It is better not to use word “we”. 

Response: We agree to the reviewer's suggestion. The sentence has been deleted. Please see Lines 464-465 in the revised manuscript.

  1. The manuscript may be recommended for publication after minor revisions.

Response: We agree to the reviewer's suggestion. Thanks very much for the reviewer’s affirmation.